# FastRLAP: A System for Learning High-Speed Driving via Deep RL and Autonomous Practicing

**Kyle Stachowicz**[†], **Dhruv Shah**[†], **Arjun Bhorkar**[†], **Ilya Kostrikov, Sergey Levine**
UC Berkeley

**Abstract:** We present a system that enables a 1/10[th]-scale autonomous car to drive at high speeds from visual observations using reinforcement learning (RL). Our system, FastRLAP (*faster lap*), trains autonomously in the real world, without human interventions, and without requiring any simulation or expert demonstrations. FastRLAP integrates several components to facilitate the learning process: we initialize low-dimensional visual representations from a similar reinforcement learning objective applied to a large offline navigation dataset from *other* robots, providing a navigation-relevant representation. Given a series of checkpoints representing a driving course, we then use sample-efficient online RL to learn a fast driving policy, resetting automatically on collision or failure. Perhaps surprisingly, our system can learn to drive over a variety of racing courses with less than 20 minutes of online training. The resulting policies exhibit emergent aggressive driving skills, such as timing braking and acceleration around turns and avoiding areas which impede the robot's motion, approaching the performance of a human driver using a similar first-person interface over the course of training.

**Keywords:** reinforcement learning, offroad driving, vision-based navigation

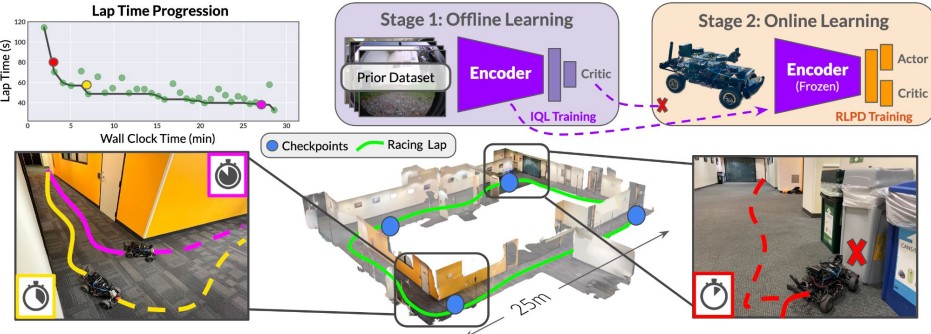

**Figure 1: Fast reinforcement learning via autonomous practicing.** By pre-training to learn task-relevant visual features (Stage 1), and deploying our autonomous practicing framework for continuous online improvement (Stage 2), the robot can autonomously navigate between sparse checkpoints (blue), recover from collisions (red) and improve its driving behavior to maximize speed (yellow → magenta). FastRLAP learns fast driving policies in as little as 20 minutes. Videos available at https://sites.google.com/view/fastrlap.

## 1 Introduction

High-speed vision-based navigation presents a range of challenges, requiring a policy that can account for both the vehicle's dynamics and interactions with the terrain and obstacles (Fig. 1). Learning-based methods offer a particularly appealing approach to such challenges, as they can in principle capture arbitrary high-performance driving behaviors while accounting for visual indicators. Some prior work has approached similar problems via imitation learning, acquiring end-to-end skills from expert demonstrations [1, 2]. However, if we aim to maximize performance, we might

7th Conference on Robot Learning (CoRL 2023), Atlanta, USA.

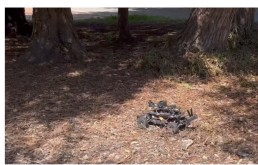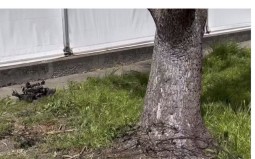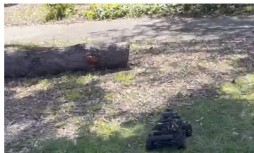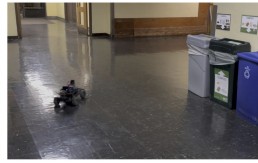

**Figure 2:** FastRLAP learns fast driving policies for a 1/10[th]-scale vehicle operating in diverse indoor and outdoor environments with challenging terrain in tens of minutes, using offline pre-training and online RL.

instead prefer to directly adapt the driving strategy to the vehicle *autonomously*. In principle, reinforcement learning allows an agent to continually improve based on its experience, as shown previously in board games and robot manipulation where RL can even exceed human performance [3–6].

However, in practice, learning autonomous navigation with RL presents major challenges. Because we cannot reset the system to a random state, the learning process is highly dependent on the system's ability to continually reach new states without human intervention. Instead, the RL-based system should train without supervision, while smoothly recovering from failures or collisions. Furthermore, directly learning from high-dimensional observations in the real world can be prohibitively slow. Because features are learned from a very weak signal (reward), RL often requires a huge number of interactions with the environment to learn a robust policy. Alternatively, we could learn entirely from offline data [7, 8], but this yields suboptimal policies when the desired behavior (aggressive driving) is not included in the dataset (low-speed navigation). The goal of this paper is to address these challenges and understand how RL can be applied to learn high-speed driving from vision in the real world. We design a system for **Fast Reinforcement Learning via Autonomous Practicing** (**FastRLAP**) which mitigates the sample complexity challenges by first learning a low-dimensional representation of driving-related features such as free space and obstacles from offline data, and then applying online RL to these features to learn a fast driving policy. The online RL phase proceeds autonomously, automatically recovering from failures and improving with each lap.

We demonstrate FastRLAP in challenging environments on a custom 1/10[th]-scale RC car modified for real-world online RL. FastRLAP can autonomously practice and learn aggressive maneuvers over time, improving by up to 40% over the demonstration lap and achieving performance close to a human expert. Notably, the online training phase typically takes less than 20 minutes (and as little as 5 minutes), depending on the size of the environment. During this time, the robot learns complex maneuvers such as drifting, avoiding low-speed or bumpy areas, and maintaining a racing line, without requiring high-speed human demonstration or explicit reward for these behaviors. The training requires no human interventions and is fully autonomous. To the best of our knowledge, FastRLAP is the first instantiation of a vision-based mobile robotic system that uses model-free RL to autonomously practice high-speed driving maneuvers and improve online in the real world.

## 2 Related Work

Leveraging prior data to bootstrap online learning has been widely studied in the context of supervised learning [9], representation learning [10], continual learning [11–15], and RL [8]. Offline RL in particular has proven

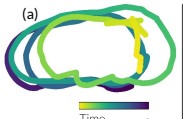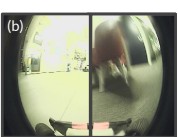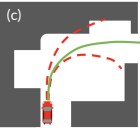

**Figure 3:** High-speed visual navigation faces challenges including: (a) noisy odometry and localization, (b) overexposure and motion blur, and (c) terrain-dependent over-/under-steer.

powerful due to its ability to directly learn policies from large datasets, which can be fine-tuned through online interaction [16–19]. This has enabled a variety of robotic systems leveraging a combination of offline data and online interaction to perform real-world manipulation tasks [20–22], typically in controlled spaces in which the offline data consists of many high-performance demonstrations with the same robot in the target environment. In contrast, FastRLAP operates with only low-performance (slow) data primarily from other robots and environments, and the majority of behaviors in the resultant policy do not appear in the original dataset.

Existing approaches for learning high-speed driving typically rely either on highly accurate position information to define states [23–26], localize visual observations relative to a high-fidelity *global*

*map* [27, 28], or operate via behavioral cloning against some privileged expert [29]. This is prohibitive in unstructured environments, where (i) onboard state estimates can be highly inaccurate, and (ii) generating a high-fidelity map is difficult or impossible. FastRLAP learns high-speed driving directly from vision, and improves its behavior by self-practice without using privileged state.

Prior success in learning visual navigation policies typically requires large-scale simulated data [30–33], passive data [34], human interventions [35, 36] or real-world data from other robots [37]. While these modalities (in particular, simulation) are typically used to overcome the high number of samples often required by RL algorithms, we demonstrate that it is possible to train such a policy in reasonable time with only real-world interactions, opening the door to policies reflecting complex relationships between vision, dynamics, and terrain (Fig. 3) that might be difficult to simulate.

Several works have studied autonomous real-world RL via safety or reset-free training [38–43] with applications in robotic manipulation, locomotion, and mobility [20, 44–46]. We draw inspiration from these works to build a high-speed navigation system that uses a finite state machine to practice driving around a circuit. FastRLAP can drive diverse courses 100+ meters in length and continually improves its performance over the course of minutes rather than hours or days.

## 3 Autonomous Practicing with RL

The objective of our high-speed visual navigation task is to drive through a race course, defined by a sequence of position *checkpoints* $\{c_i\}$, in the minimum possible time. We assume access to two sources of offline data, neither of which contains the desired high-speed behavior: a large-scale dataset representing common navigation behaviors executed on a different robot, and a small dataset including a single lap around the course at low speed. Our system aims to enable efficient end-to-end RL in the real world. FastRLAP has three components (see Fig. 1): a high-level finite state machine (FSM) for autonomous practicing (shown in blue), a representation of visual observations learned via offline RL (purple), and a sample-efficient RL algorithm for online learning (orange).

### 3.1 Problem formulation

We frame this task as a Markov decision process $\mathcal{M}(\mathcal{S}, \mathcal{A}, p, r)$ with state $(V, v, \omega, \alpha, g, a_{\text{prev}}) \in \mathcal{S}$. Here, $V \in \mathbb{R}^{128 \times 128 \times 9}$ is a sequence of 3 RGB images; $v, \omega, \alpha \in \mathbb{R}^3$ denote the robot's body-frame linear velocity, angular velocity, and linear acceleration; the goal $g$ is a body-frame vector to the next checkpoint, expressed as a unit vector and a distance; $a_{\text{prev}}$ is the previous action.

In order to align the visual representations learned offline with those most useful for the online task, both offline learning and online training phases are structured to maximize the same reward, ensuring optimal transfer between the two settings. To this end, we define the reward as the weighted sum of three components: **speed-made-good**, which is the dot product of the current velocity with the unit vector pointing towards the current goal; a **collision penalty** proportional to the magnitude of the collision (measured by lateral acceleration) applied only when a collision is detected; and a fixed **stuck penalty** applied whenever the robot is determined to be "stuck" by the practicing system.

### 3.2 Autonomous Practicing and Goal Checkpoint Selection

In the autonomous learning setting, the robot is expected to learn in the environment without any episodic resets or human interventions. Early in training, the policy may reach irrecoverable states, such as collisions, or otherwise become stuck. Without a reset, the learning algorithm may fail due to collapse in the state distribution [41]. To overcome this, we use a simple FSM that switches between a simple collision recovery policy and the learned policy.

When the RL policy reaches a checkpoint, the FSM selects a new goal corresponding to the next checkpoint in the course sequence $\{c_i\}$, forcing the learner to practice reaching all of the checkpoint goals in sequence. The goal checkpoints $c_i$ are typically beyond line-of-sight (e.g., Fig. 1, blue), up

to 40 meters away. If the RL policy reaches an irrecoverable state (see Sec. 4), the FSM commands an automatic recovery policy to provides a "pseudo-reset."

## 3.3 Online RL Training

To maximize reward and continually improve lap times, we use off-policy RL [3, 47]. Off-policy algorithms benefit greatly from performing many training steps for each environment step, known as the update-to-data ratio (UTD): high UTD leads to efficient learning, but suffers from overfitting [48]. To overcome this limitation, we use RLPD [49], which trains an ensemble of critics to avoid catastrophic overestimation and overfitting [50] and learns quickly using a combination of online interactions and a small amount of suboptimal, *on-task* data.

We obtain this on-task data by collecting a single *slow* lap in the target environment. While this data is very limited (under a minute in most environments) and does not contain fast driving behaviors, even suboptimal demonstations can significantly accelerate online learning by avoiding critic collapse in early stages of training [49]. During online training, we sample

---

**Algorithm 1:** FastRLAP

**Data:** Navigation dataset $\mathcal{D}$, slow demo $\mathcal{B}_{\text{slow}}$
1 **Keys:** Pre-Training, Practicing, Online RL
2 **while** *Encoder is not converged* **do**
3     $s, a, s', \text{idx} \leftarrow \text{LoadData}(\mathcal{D})$
4     $g \leftarrow \text{LoadFutureData}(\mathcal{D}, \text{idx} + \text{Rand}(H))$
5     $r \leftarrow \text{ComputeReward}(s, a, g)$
6     $\text{Train}_{\text{IQL}}((s, g), a, r, (s', g))$

7 **while** *True* **do**
8     **On Robot**
9       $s \leftarrow \text{Observe}()$
10       **if** $s$ near $g$ **then**
11         $g \leftarrow \text{NextCheckpoint}(g)$
12       $r \leftarrow \text{ComputeReward}(s_{\text{prev}}, a_{\text{prev}}, g)$
13       $\text{SendToWorkstation}(s_{\text{prev}}, a_{\text{prev}}, r, s, g)$
14       $a \sim \pi(\phi(s_{\text{image}}), s_{\text{proprio}}, g)$
15       $\text{Actuate}(a)$
16       **if** *Collision* or *Stuck* **then**
17         Execute recovery policy

18     **On Workstation**
19       $\text{ReceiveFromRobot}(\mathcal{B})$
20       $b \leftarrow \text{Sample}(\mathcal{B}), b_{\text{d}} \leftarrow \text{Sample}(\mathcal{B}_{\text{slow}})$
21       $\pi, Q \leftarrow \text{Train}_{\text{RLPD}}(\pi, Q, b, b_{\text{d}})$

---

50% of each training batch from this low-speed data, interleaved with 50% of data collected online. We found this to be critical to the efficiency of our system in our evaluations (Sec. 5.1).

## 3.4 Representation Learning with Offline RL

When training image-to-action policies, end-to-end RL allows gradients from the control objective to optimize the encoder. This results in a task-specific encoder that produces features that are most relevant to the agent's task, rather than general features (e.g., features necessary for classification or video prediction). Unfortunately, training directly on full images is very computationally expensive and unacceptably reduces the UTD ratio. Ideally, we would prefer to pre-train some encoder to produce task-relevant features *offline*, and then freeze the encoder during online training.

We address this by training the encoder with offline RL on an existing large-scale dataset with a similar objective in a different setting. In particular, we use RECON [51], a large-scale navigation dataset collected by manually driving a Clearpath Jackal UGV outdoors at low speeds. This dataset contains navigation trajectories from many environments and an entirely different robot, but importantly does *not* include aggressive high-speed driving. Thus, the role of pre-training is not to teach the robot how to drive quickly, but only to extract a relevant representation to simplify the online learning problem. The high-speed driving behaviors necessary to solve the desired task must be learned through practice in the real world, building on this pre-trained foundation.

We apply goal-conditioned offline RL by selecting a 1:1 mixture of random goals and goals from the robot's future trajectory in this dataset, and use Implicit Q-Learning [52] to train a critic network (illustrated in Fig. 1, purple). We then take the learned encoder, which now encodes features relevant to the navigation task, and freeze it for training the policy and critic (orange) as illustrated in Sec. 3.3.

## 4 System Design for Online Learning

We instantiate FastRLAP on a 1/10th-scale autonomous car [53, 54]. Our system is based on a Traxxas Slash 4×4 modified to facilitate online learning. See the appendix for a full parts list.

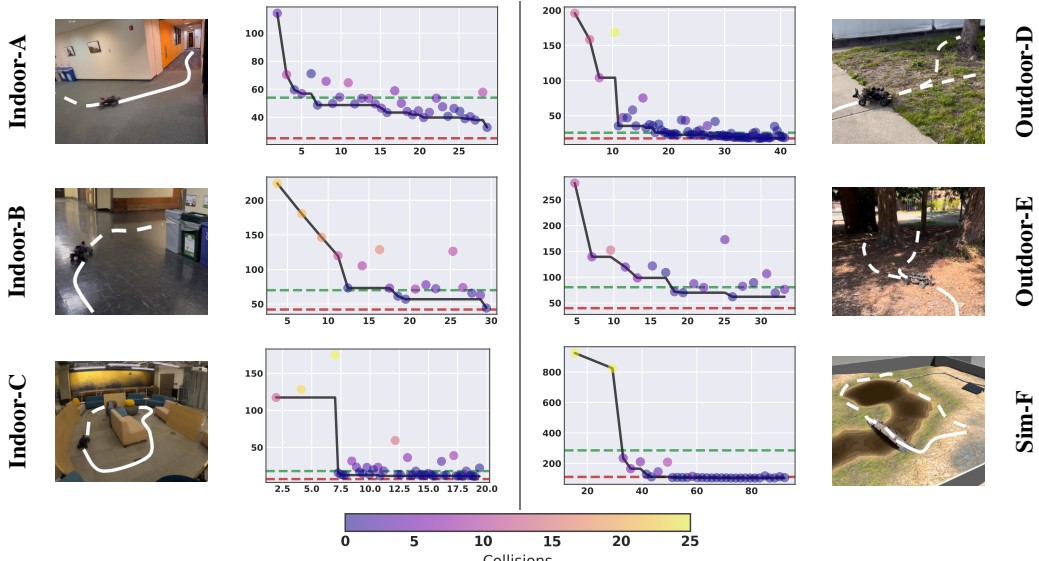

**Figure 4:** FastRLAP achieves high-speed driving in diverse environments via autonomous practicing. FastR-LAP improves lap times (best-so-far shown in black) given a slow *demo lap* (green) and achieves near-expert lap times (red) in under 40 minutes.

**Sensing:** Since our high-speed system operates directly on visual observations, we use a forward-facing fisheye camera to obtain a low-latency stream of $128 \times 128$ RGB images. The policy also depends on IMU data and motor speed as well as relative checkpoint position from a state estimator.

*Indoor state estimation:* Indoors, we use a RealSense T265 tracking camera, mounted facing the ceiling, to estimate the robot's pose and velocity.

*Outdoor state estimation:* We mount a GPS receiver onboard the robot. To estimate absolute heading $\theta$, we use an extended Kalman filter to fuse wheel odometry $v_w$, absolute GPS velocity $\vec{v}$, and angular velocity $\omega$ with dynamics $\theta_{t+1} = \theta_t + \omega \Delta t$ and measurement model $\vec{v} = (v_w \cos \theta, v_w \sin \theta)$.

**Compute:** We use an NVIDIA Jetson Orin NX for onboard compute. We process visual observations onboard using a pre-trained encoder (Sec. 3.4), and offload training to a workstation with a GTX 1080ti GPU. We implement our algorithm in JAX [55] and compile several training steps into a single function, allowing ∼800 actor-critic updates per second (vs. ∼80 without any optimizations).

**Actuation:** Standard RC motors exhibit "cogging", a stuttering behavior depending heavily on the (unobserved) rotor position. We instead use a sensored motor to provide closed-loop sequencing.

**Action space:** The action space consists of a steering angle and a target speed, which is limited to 4.5m/s across most of our experiments to avoid damaging the robot. This limitation is addressed in Sec. 5.2 to safely learn a policy driving at higher speeds. To ensure smooth actions, (i) we use a *shifted* $\tanh$ linearized around $a_{\text{prev}}$ to constrain the action to $[a_{\text{prev}} - \delta, a_{\text{prev}} + \delta]$, and (ii) we append the previous action to the observation. See the appendix for details.

**Detecting blocked states:** The state machine triggers a pseudo-reset and delivers negative reward when the robot collides, detected by high lateral acceleration, or is not moving for 3 seconds.

## 5  Faster Lap Times with FastRLAP

In this section we present an experimental evaluation of FastRLAP in a variety of real-world and simulated environments. We consider several metrics to analyze the peak performance, as well as cumulative metrics during practice. The time-to-first-lap (T2F) represents the time taken to complete the first full collision-free lap, starting from scratch. We track the best lap time achieved during training as well as the median time of last five laps completed to capture the converged behavior.

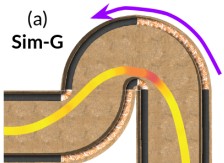 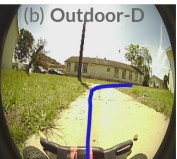 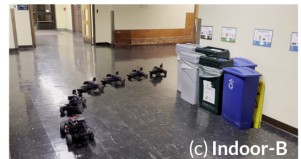 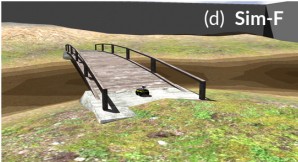

**Figure 5: Emergent behaviors with FastRLAP.** Maximizing speed with RL results in a "racing line", braking as late as possible to maintain speed in and out of tight corners (a) and drifting on slick surfaces (c). Outdoors, tall grass slows the robot's motion, promoting driving on paths (b). In **Sim-F**, the robot infers that the bridge is faster than driving through mud via visual correlation (d).

| Env. | T2F (min) | Lap Times (s) | | | | # Collisions Median |
|---|---|---|---|---|---|---|
| | | Best | Median | Demo | Expert[†] | |
| **Indoor-A** | 4.07 | 32.7 | 39.0 | 54 | 25 | 0 |
| **Indoor-B** | 12.41 | 44.2 | 65.7 | 70 | 43 | 3 |
| **Indoor-C** | 7.27 | 10.9 | 11.7 | 17 | 7 | 0 |
| **Outdoor-D** | 11.04 | 17.1 | 22.7 | 43 | 18 | 0 |
| **Outdoor-E** | 19.29 | 62.1 | 94.0 | 160 | 40 | 3 |
| **Sim-F** | 8.11 | 104.1 | 107.0 | 286 | 112 | 0 |
| **Sim-G** | 41.54 | 18.0 | 18.1 | 36 | 19 | 0 |
| **Outdoor-D** (Schedule) | 21.5 | 13.1 | 23.4 | 43 | 18 | 0 |

**Table 1: Summary of experiments:** FastRLAP rapidly learns fast driving policies in environments of varying difficulty levels, improving over the *demo lap* by over 40% and achieving lap times within 5% of the expert, using only egocentric observations.

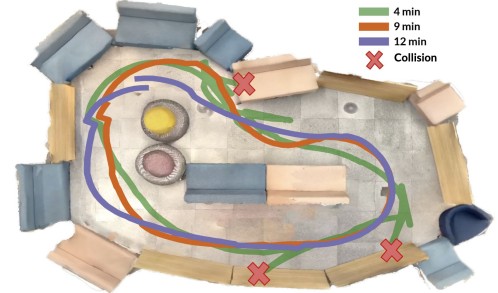

**Figure 6:** Sample trajectories of FastRLAP practicing in **Indoor-C**. FastRLAP recovers from collision (green) and learns collision-free navigation (orange). Maximizing speed, FastRLAP discovers a smooth racing line (purple). *The 3D scan is shown only for illustration.*

Additionally, we list the median collisions in the last five laps to capture safety. To contextualize our results, we provide timing for laps driven in each environment by human drivers watching the robot from a third-person view ("Human Expert"), as well as the duration of the "slow demo" lap.

We used the same hyperparameters (network architecture, learning rate, etc.) for all experiments, both in the real-world and simulation. See the appendix for a full list of hyperparameter values, detailed laptime plots for all baselines, and additional qualitative analysis.

## 5.1 Real-World Deployment

We deploy FastRLAP in several diverse environments to demonstrate autonomous practicing. Before training, we manually drive the robot around the course for a slow lap to define the rough layout of the track. This lap is used in two ways: (i) to generate a sequence of sparse checkpoints $\{c_i\}_{i=1}^{n_c}$ for the practicing FSM described in Sec. 3.2, and (ii) to provide a low-speed demonstration for off-policy actor-critic updates as described in Sec. 3.3.

We test in five real-world environments: three indoor and two outdoor, labeled **A-E**, shown in Fig. 4. We also test in two simulated environments, **Sim-F** and **Sim-G**. Environments include challenging features such as large scale (**Indoor-C**, **Outdoor-E**, **Sim-F**), tight or cluttered navigation (**Indoor-B**, **Outdoor-E**, **Sim-G**), and highly terrain-dependent speed that must be inferred by correlating visual observations to proprioceptive speed measurements during training (**Outdoor-D**, **Outdoor-E**, **Sim-F**). All environments are described in detail in the appendix.

Table 1 and Fig. 4 summarize the performance of our system in these environments. FastRLAP is able to consistently improve over the low-speed demonstration lap in a handful of laps, and nearly match human performance in **Indoor-B** and **Outdoor-D** in 30 minutes of real-world practice, without any human interventions. As training progresses, the achieved lap times continue to decrease, with the robot's path becoming smoother as a secondary effect of optimizing speed Fig. 6.

**Emergent behaviors**: Maximizing the reward for reaching checkpoints quickly leads to several emergent behaviors. The system learns a racing line, optimizing speed through corners and chicanes (a). In Fig. 5(a), the robot maintains speed through the apex of a tight corner, braking sharply to turn and accelerating out to minimize driving time. On a low-friction surface (c), the policy over-steers slightly, achieving fast rotation without braking. Outdoors, the learned policy prefers smooth, high-traction areas on and around concrete paths (b), avoiding tall grass that hinders motion.

## 5.2 Even Faster Laps: Scheduling Speed Limits

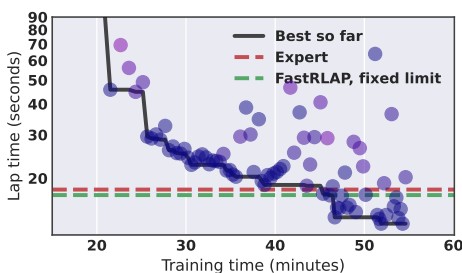

The action space described in Sec. 4 allows limits to be adjusted arbitrarily during training. Leveraging this property, we push the limits of FastRLAP in **Outdoor-D**, increasing the action bounds linearly over time from an initial maximum speed of 2.5m/s to a final speed of 6.75m/s. By increasing the maximum commanded speed smoothly from a slow initial limit, FastRLAP first learns basic behavior in the low-speed setting that then transfers to high-speed driving, without causing crashes at high speeds.

**Figure 7:** Lap time progression with scheduled increases in the speed limit. Note log scale.

We see that in the **Outdoor-D** setting FastRLAP is able to learn a much more aggressive policy than with the original limits, both quantitatively (Fig. 7) and qualitatively (see the videos on our website).

## 5.3 Comparative Analysis

We compare the performance of FastRLAP against several baselines and ablations in **Indoor-C** and **Sim-G** to demonstrate the importance of each of the components of our method: pre-trained visual representations, online RL starting from a slow demo lap, and autonomous recovery behaviors to handle the reset-free environment. Specifically, we consider the following variations:

**No Demo Lap:** Ablate the demonstration lap and use only online data.

**No Pre-Training:** Ablate pre-training and use DrQ [56] to learn the encoder from scratch.

**No Pseudo-Resets:** Ablate the scripted pseudo-resets, requiring the robot to learn recovery behavior.

**ImageNet Pre-Training:** Ablate task-specific pre-training and instead train the encoder for image classification on ImageNet [57, 58] (with the same encoder structure).

**DINOv2 Pre-Training:** Uses DINOv2 ViT-S [59], a self-supervised vision transformer, as the encoder in place of RL pre-training. The much larger model introduces roughly 40ms of actor latency.

**Offline RL:** Ablate online learning and use a policy trained purely offline with 15 minutes of *expert* data from the replay buffer of a successful run of FastRLAP using IQL [52].

**State-Based:** Replaces visual observations with *privileged* state (absolute $x, y, \theta$ from VIO).

In both **Indoor-C** and **Sim-G**, FastRLAP outperforms the ablation with no demo lap in both time-to-first lap (T2F) and best lap time while causing fewer collisions (Tab. 2). The demo lap helps the robot make progress early in training, enabling broad state coverage and better final performance. Removing pseudo-resets causes the robot to become stuck, causing similarly poor performance.

While initializing FastRLAP with a general-purpose pre-trained visual encoder (DINOv2 and ImageNet) gives reasonable performance in simulation, its performance is comparatively poor in the real-world **Indoor-C**. This suggests that while general-purpose visual features are sufficient for low speeds, high-speed navigation requires task-specific features. Training the encoder online achieves good asymptotic performance, but takes a long time to complete its first collision-free lap and improves relatively slowly due to a reduced UTD ratio (Fig. 8). Our approach also outperforms the variant with access to privileged state information, suggesting that the pre-trained features generalize better than a simple localization estimate.

| Name | Indoor-C | | | | Sim-G | | | |
|------|----------|------|--------|------------|-----|------|--------|------------|
| | T2F | Best | Median | Collisions | T2F | Best | Median | Collisions |
| State-Based | 11.07 | 12.7 | 18.8 | 4 | 9.5±2.0 | 21.1±1.7 | 26.2 | **0** |
| No Demo Lap | 14.64 | 16.0 | 62.6 | 12 | 9.8 ± 2.2 | 20.5 ± 0.9 | 22.2 | **0** |
| Offline RL [60] | ∞ | – | – | – | – | – | – | – |
| No Pre-Training | 10.34 | 12.7 | 20.0 | 1 | 10.3 ± 2.2 | **19.3±1.3** | 18.4 | **0** |
| ImageNet Encoder | 10.05 | 19.7 | 29.7 | 1 | 8.2 ± 1.4 | 21.0 ± 2.7 | 22.1 | **0** |
| DINOv2 Encoder [59] | 16.09 | 17.0 | 34.8 | 4 | 8.6 ± 1.7 | 20.6 ± 1.3 | 25.6 | **0** |
| FastRLAP (Ours) | **7.27** | **10.9** | **13.3** | **0** | **6.9±0.9** | **19.3±0.1** | **18.1** | **0** |
| Human FPV | – | 11.1 | 14.4 | 2 | – | 18.6 | 18.9 | 0 |
| Human Oracle[†] | – | 7.3 | 8.8 | 0 | – | – | – | – |

**Table 2: Comparing to baselines**: In real and simulated environments, FastRLAP has faster time to first lap (T2F), best/median lap times, and median collisions, and achieves near-human performance. Offline RL does not complete a collision-free lap. T2F listed in minutes; other times in seconds; lower is better for all metrics. Simulation results are reported as mean±std. dev over 3 seeds.

## 6 Discussion

We presented a system for learning high-speed driving with reinforcement learning from rich observations, practicing autonomously in the real world. Our approach uses representations

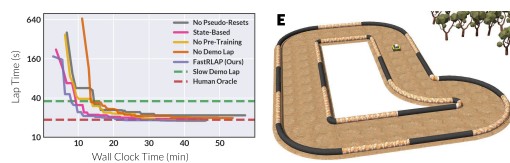

**Figure 8:** Lap times for baselines in **Sim-G**.

from prior data to initialize the policy, followed by sample-efficient online RL and a checkpoint-based navigation strategy to recover autonomously from collisions and continue practicing. Although deep RL is often believed to be inefficient and difficult to use in the real world, we demonstrate that with appropriate pre-training and system design it is possible to learn effective driving strategies in less than 20 minutes of real-world training. This result may seem quite surprising when viewed in contrast to prior work that uses simulated data [30] or hundreds of hours of training [61], and it provides strong validation that deep RL, in conjunction with task-specific pre-training and approximate resets, can indeed be a viable tool for learning real-world policies from raw images.

A qualitative investigation of the policies learned by our system also reveals interesting emergent behavior. Although we bootstrap training with prior data (in other domains and from other robots) and a single slow demonstration lap, the learned policies exhibit behaviors deviating significantly from the dataset including drifting, selecting for high-speed terrain, and maintaining a racing line. Thus, the online RL process not only robustifies existing behavior, as observed in prior work [21], but also acquires new emergent behaviors by building on the foundation established by the prior data. Our ablations establish the importance of *task-relevant* pre-training, supporting the notion that representations learned from diverse robot navigation data serve as an effective foundation for downstream skill learning — much as pre-training enables efficient fine-tuning for vision and NLP [62, 63].

**Limitations and future work:** While our system enables highly effective image-based driving, it does have several limitations. First, the current implementation requires a coarse state estimator to provide a vector to the next checkpoint. This could be addressed in future work by specifying future goals in another format, such as images [64]. Second, our system does not explicitly account for *safety* during the training process: the agent will learn to avoid collisions because they lead to task failure, but high-speed collisions during training could cause damage. Future work could include a conservative or risk-aware formulation to counteract this effect. Nevertheless, we believe that our work represents a step towards RL-based systems that can autonomously learn highly performant navigation skills in a wide range of domains.

## Acknowledgments

This research was partially supported by DARPA RACER, ARL DCIST CRA W911NF-17-2-0181, the National Science Foundation through IIS-2150826, and the Office of Naval Research. The authors would like to thank Alejandro Escontrela, Noriaki Hirose, and Philippe Hansen-Estruch, for their help with running experiments and providing baseline implementations.

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
