# OpenReview forum: "FastRLAP: A System for Learning High-Speed Driving via Deep RL and Autonomous Practicing"
_robot-learning.org/CoRL/2023/Conference — CoRL 2023 Poster_

### Official Review · Reviewer_q79b · 2023-07-17

**Confidence:** 4
**Originality:** Good
**Technical Quality:** Excellent
**Clarity Of Presentation:** Excellent
**Impact:** 3

**Recommendation:**

Weak Accept: I recommend accepting the paper, but will not argue for my recommendation if the majority of other reviewers have a different opinion.

**Review:**

The paper is clear in stating its contribution, and is generally attacking a challenging problem.  Because the paper is largely concerned with the practical aspects of RL in racing environments, the novelty is largely in the implementation details.  However, the extensive evaluation conducted in indoor, outdoor and simulated environments greatly strengthen its contributions.  Nonetheless, there are flaws in the paper that should be addressed.

The authors could clarify what properties the offline dataset is needed to freeze the encoder for online use.  It seems that the dataset would need to have complete coverage of the entire environment, and the online-trained policy could catastrophically suffer if it encountered a new state not included in the offline data.  What are the consequences of updating or fine-tuning the encoder online?  The authors should explain such design rationales, and shoud consider including alternative amounts of data as ablations – in the extreme, no small-scale dataset, and different training splits between online and low-speed offline data.

I wonder why, in the offline step, only the encoder is carried forward into the online step, and why it is useful to collect large-scale offline data using a different robot.  This seems overly restrictive.  In offline RL, it seems natural that the actor policy could be warm-started using the offline one, given that they are both learned from the same reward in the same environment. Relatedly, if there are two robots, does the encoder suffer from any camera pose or other perception differences when going from one robot to the next?

The “state-based” ablation seems a bit of an unfair comparison.  The authors seem to compare against a policy trained on instantaneous pose observations.  Other approaches that use proprioceptive observations assume awareness of a local map (obstacles, road bounds, a linear fit of the checkpoints, etc), and including velocity as an observed state would seem more sensible and straightforward.  The authors should consider updating this ablation to include such information, or discuss why this was not used.

The nomenclature used in Algorithm 1 are not properly defined.  For instance $\mathcal{B}$, $b_d$, $s_{slow}$, $s_{prev}$, etc. are not defined at all.  In Table 2, the bolding of the 18.0 for FastRLAP under “Sim-G”, “Best” seems incorrect, as No Pretraining seems to outperform it.


**Quality Of The Limitations Section:**

Limitations are addressed clearly

**Questions For Rebuttal:**

Could the authors explain the motivation for the data choices: in particular why the choice of training the actor from scratch and freezing the encoder during the online phase make sense in this setting?

**Robotics Focus:**

Sufficient demonstration on hardware

**Summary Of Paper:**

The paper addresses the question of training a vision-based policy for racing by blending offline and online data.  The offline pre-training phase is used primarily in learning a vision representation, while the online fine tuning phase allows training and adapting a policy.   Overall, the contribution of the paper lies in addressing some practical problems of sample efficiency in offline RL applied to the racing domain in various realistic environments.  The procedure assumes access to, and observability of, a series of checkpoints along the course, a set of recovery policies, and a set of offline data: one with a diverse set of views along the course (not necessarily taken from the target robot), and another with the target robot taken at slow speed.

**Summary Of Recommendation:**

Overall, the paper is well written, and the approach itself is sound.  The paper addresses a long-standing issue in RL in vision-based planning, and narrows the gap in applying offline RL to vehicle planning and control in real-world settings.  While the factorization of the problem into pre-training and online fine-tuning is not a new idea, the evaluation in a diverse set of experiments makes for a strong contribution.  I lean towards acceptance.

---

### Official Review · Reviewer_gJxi · 2023-07-17

**Confidence:** 3
**Originality:** Good
**Technical Quality:** Very Good
**Clarity Of Presentation:** Excellent
**Impact:** 3

**Recommendation:**

Weak Accept: I recommend accepting the paper, but will not argue for my recommendation if the majority of other reviewers have a different opinion.

**Review:**

This paper is well written and the accompanying video is impressive. The paper is also practical: I appreciate the switching between a recovery policy and a learned policy. I also appreciate the inclusion of ablation studies.

I am not an expert in the precise area of focus of this paper. My first thought on reading this paper was that it seemed like there might be little novelty to it (especially on reading the last sentence of the introduction, "FastRLAP is the first instantiation of a vision-based mobile robotic system that uses model-free RL to autonomously practice high-speed driving maneuvers and improve online in the real world"; this seemed unbelievable). However, I did a small literature search, and I didn't find anything exactly comparable. Two papers which seem useful:
- 1, Xiao, Xuesu, et al. "Autonomous Ground Navigation in Highly Constrained Spaces: Lessons Learned From the Benchmark Autonomous Robot Navigation Challenge at ICRA 2022 [Competitions]." IEEE Robotics & Automation Magazine 29.4 (2022): 148-156.
- 2, This paper was put on ArXiv after the author’s CoRL submission, so I am only listing it as a potential resource and not asking for a comparison: Shah, Dhruv, et al. "ViNT: A Foundation Model for Visual Navigation." arXiv preprint arXiv:2306.14846 (2023).
Another related work is GT-Sophy, though of course that takes place entirely in simulation.

The paper's results do not list any kind of error bounds - i.e., it would be nice to have the system evaluated ~10 times and provide the variance of the completion times to help us assess whether the FastRLAP is a significant improvement. For example, the No Pre-Training numbers in Table 2 look relatively close.

I find the related work section of the paper to be weak; it does not help me position this work in the larger research context. If this paper is to be accepted, I would ask the authors to rewrite this section with this goal of positioning the paper in the larger research context in mind. The related work should, in my opinion, help an expert in an adjacent research area assess where to learn more about this specific research problem. Broad citations to supervised learning, representation learning, continual learning, etc do not help with this. For this paper, I would like to know: is this reset mechanism and policy switching approach new (how do citations 38-43 relate, and what exactly is relevant about them)? Are there different settings where others have successfully trained a policy first with offline data and then with a small number of imperfect online samples? What is unique about this racing problem? Etc.

This paper mentioned that it uses offline data to first learn a navigation-ready representation, but exactly what this data is composed of is not stated until Section 3 (and not precisely explained until Section 3.4). The precise composition of this data should be discussed in the introduction.

As written, the codebase is uncommented and very hard to reuse, so work would need to be done to improve this resource. I am concerned about reproducibility of this paper. This is always hard with robotics settings, but a well-commented codebase would significantly help.

The graphs in figure 4 need more explanation and axis labels.

Minor comments:
3.1 should define each term of the MDP and explain how they are modeled for this problem. Also missing the mention of the discount factor.

Figure 4's horizontal order should be image-A graph-A | image-B graph-B (instead of current order image-A graph-A | graph-B image-B).

Line 257 typo "before before"

Table 2 has an incorrectly highlighted value (FastRLAP, Best for Sim-G).


**Quality Of The Limitations Section:**

Limitations are addressed clearly

**Questions For Rebuttal:**

How was the reward function designed? Why was the reward function structured this way?

Line 208 refers to 'maximizing the reward for reaching checkpoints quickly' but this is not part of the reward fn described earlier? (It is implicit in the speed-made-good term, but this is weighted against two other terms. It is also implicit, to some extent, in the discount factor.)



**Robotics Focus:**

Sufficient demonstration on hardware

**Summary Of Paper:**

This paper describes a method for learning fast navigation through a combination of offline data (consisting of previously-released data collected by another robot in other train environments, as well as a slow lap through the test environment) and online learning. The paper evaluates this method by demonstrating its ability to learn these navigation tasks and by ablating each component of the method.


**Summary Of Recommendation:**

I am unfortunately not an expert in the navigation subfield, so I will weight my review lesser than other reviewers with more direct expertise.

I found this paper to be well written and reasonable in its approach. I have some concerns about reproducibility and some requests for edits (e.g., a new related work section). On balance, I think the work is well suited to the CoRL audience and is sufficiently polished to recommend its acceptance.

---

### Official Review · Reviewer_ieSQ · 2023-07-20

**Confidence:** 5
**Originality:** Good
**Technical Quality:** Very Good
**Clarity Of Presentation:** Very Good
**Impact:** 3

**Recommendation:**

Weak Accept: I recommend accepting the paper, but will not argue for my recommendation if the majority of other reviewers have a different opinion.

**Review:**

The paper is well written and easy to follow.

# Strengths

- The use of an FSM to provide intermediate checkpoints and a collision-recovery subroutine to facilitate “in the wild” reinforcement learning in a sample-efficient, reset-free manner is impressive, novel as far as I’m aware, and brings the field closer to making RL more reliable in real-world settings.

- The ablations are comprehensive and have convinced me that the authors’ design decisions wrt encoder pretraining, the use of “low quality” demonstrations to facilitate learning, and the FSM are important aspects of the algorithm.

- The learned policies demonstrate real-world driving capability in indoor and outdoor environments, under different terrains, lighting and motion blur conditions, and surfaces.

# Weaknesses

- The algorithm learning emergent, aggressive behaviors not present in the initial dataset, while impressive, is not entirely as novel as the authors make it seem, and should be situated within the broader context of prior methods. Broadly speaking, the training paradigm laid out by the authors can be described as "offline pre-training plus online fine-tuning" or "imitation plus reinforcement learning". The former has been studied in methods such as [1, 2], and the later has been documented and well-studied within the context of autonomous driving in papers such as [3]. The question that then arises is, how does this approach compare to these prior methods and, how is the current approach situated in the broader space of offline pretraining with online fine-tuning? The authors could have discussed this in greater detail in the paper.

- (Section 6) *This result may seem quite surprising when viewed in contrast to prior work that uses simulated data [30], or hundreds of hours of training [61],   and we believe it provides strong validation that deep RL can indeed be a viable tool for learning real-world policies even from raw images*. The authors make misleading claims that RL itself can be used to train real world policies in under 20 minutes, whereas in reality the FSM, prior demonstrations, and pre-training of an encoder on a task-specific dataset are all employed to address the well-known weaknesses of RL without which the algorithm would work poorly, if at all, as demonstrated by the ablations. Specifically, the FSM and demonstrations provide sufficient exploration, help the RL agent recover from terminal states, and provide an initial manifold of driving policies for the RL component that helps stabliize training. To be clear, I am not against utilizing classical or hand-crafted systems to aid RL in addressing its weak points, but the authors should be honest that their results stem from a combination of classical and learned components, rather than claiming RL itself yielded these results.

[1] Nakamoto, Mitsuhiko, et al. "Cal-ql: Calibrated offline rl pre-training for efficient online fine-tuning." arXiv preprint arXiv:2303.05479 (2023).

[2] Walke, Homer Rich, et al. "Don’t start from scratch: Leveraging prior data to automate robotic reinforcement learning." Conference on Robot Learning. PMLR, 2023.

[3] Lu, Yiren, et al. "Imitation is not enough: Robustifying imitation with reinforcement learning for challenging driving scenarios." arXiv preprint arXiv:2212.11419 (2022).

**Quality Of The Limitations Section:**

Limitations are addressed clearly

**Questions For Rebuttal:**

n/a

**Robotics Focus:**

Sufficient demonstration on hardware

**Summary Of Paper:**

The authors contributions can be summarized as follows:

- Offline pretraining of an image encoder on a task-relevant, autonomous driving dataset

- Finite state machine that provides high level checkpoints to facilitate online learning and a collision-recovery subroutine that facilitates a reset-free learning paradigm

- “low quality” initial demonstrations that stabilize and accelerate the learning process combined with online data to learn emergent behaviors and driving policies that close the gap with expert-driver results.


**Summary Of Recommendation:**

The authors propose a system in which classical components such as FSM are combined with a modern offline-RL algorithm in the "offline-pretraining online fine-tuning" paradigm. The system shows generalization capability to various indoor and outdoor environments, and the ability to learn autonomously in a reset-free manner, both of which are impressive. However, the novelty aspect of the contributions seems somewhat limited, especially when placed with in the broader context of the "offline pretraining + online fine-tuning" or IL + RL literature.

---

### Official Review · Reviewer_1JBu · 2023-07-23

**Confidence:** 3
**Originality:** Very Good
**Technical Quality:** Very Good
**Clarity Of Presentation:** Very Good
**Impact:** 4

**Recommendation:**

Strong Accept: I recommend accepting the paper and will argue for my recommendation even if other reviewers hold a different opinion.

**Review:**

Overall, I found the paper very interesting, an exciting application of autonomous RL, and the experiments quite thorough.

Main Comments
- Especially since the same architecture and hyperparameters were used throughout, it would be valuable to have that information in the main paper.
- The discussion of emergent aggressive driving behavior could be more rigorous. Perhaps some quantification of how often such behavior appears in the offline data (is it really 0?), or a more precise description of how to measure such behavior, would be useful.
- It would have been great if there could be additional insights into the enviornment similarity required between the pre-training data and the test domain. While the ImageNet encoding baseline is interesting, I think ImageNet is too far away visually, and this baseline muddles both change in environment and change in training task (image classification vs. IQL). It would be great to understand why these baseliens do well in Sim but not Real more as well.
- Was the Offline RL baseline properly trained? What if more data was used? More discussion on the failure modes of this baseline would be really useful, as it is the baseline that directly shows what happens if we relax the assumption that we have no expert demonstrations.
-  I think the paper could benefit from a more prescriptive limitations section, including describing (i) why specifying goals as images was not already done in the work, (ii) describing precisely how safety considerations could be integrated into the current method, and (iii) further description of what other domains the author thinks their method could work on (e.g. what are other examples where there is sufficient offline data to use for pre-training, but the desired behavior is unlikely to be included in this set?).

Very Minor
- In 5.3, it seems like Real-C and Indoor-C are used to refer to the same thing.
- Line 48: "Without *us* explicitly demonstrating these behaviors"? On first read, it seems like it states the robot is not demonstrating these behaviors.

**Quality Of The Limitations Section:**

Additional details required

**Questions For Rebuttal:**


- How familiar were the human experts with the task? Were they "experts" at the task, or just random human drivers? Given the specialized nature of the task, I was actually surprised that they outperformed the current method -- what were some qualitative differences between FastRLAP and human experts?

**Robotics Focus:**

Sufficient demonstration on hardware

**Summary Of Paper:**

The authors introduce FastRLAP, a method for making online autonomous reinforcement learning (with images) more performant by pre-training on low-quality offline data that is task-relevant, but not in the exact domain / with the desired target behavior. They specifically apply this to task of fast, aggressive racetrack driving, and show experimental results both in simulation and in indoor and outdoor real world environments.

**Summary Of Recommendation:**

Overall, I really liked this paper - the experiments and ablations were very thorough, the results impressive, and the method well-motivated and clearly described.

---

### Decision · Program_Chairs · 2023-08-30

**Decision:**

Accept (Poster)

**Comment:**

## Summary of Paper
The paper proposes a method to speed up RL by using "low quality" initial data. The approach is evaluated in a racing task both in simulation and the real world.

## Summary of Reviews
The reviewers found the paper interesting and well written. The experiments are extensive and clearly show the benefits of the method. Some inaccurate claims were pointed out, some clarifications and improvements int he writing are needed, the embedding in the state-of-the-art should be improved.

## Influence of Rebuttal
The rebuttal addressed all major concerns of the reviewers. Reviewer ieSQ still has some concerns about the positioning of the paper.

## ToDos for Final Version
Please implement all promised improvements (limitations section, adjusted claims, error-bars on simulation results, etc.) and include the clarifications.